# Synthesizing a Water-Soluble Polymeric Nitrification Inhibitor with Novel Soil-Loosening Ability

**DOI:** 10.3390/polym16010107

**Published:** 2023-12-29

**Authors:** Yu Liu, Hui Gao, Shanshan Liu, Jinrong Li, Fangong Kong

**Affiliations:** 1State Key Laboratory of Biobased Material and Green Papermaking, Qilu University of Technology, Shandong Academy of Sciences, Jinan 250353, Chinakfgwsj1566@163.com (F.K.); 2Key Laboratory of Paper Science and Technology of Ministry of Education, Faculty of Light Industry, Qilu University of Technology, Shandong Academy of Sciences, Jinan 250353, China; 3School of Mechanical Engineering, Hebei University of Technology, Tianjin 300401, China; lijinrong@hebut.edu.cn

**Keywords:** nitrification inhibitor, acrylamide, bio-based acrylic acid, soil-loosening, nitrogen fixation, phosphorous-solubilizing

## Abstract

Nitrification inhibitor is essential for increasing the nitrogen utilization efficiency of agricultural plants, thus reducing environmental pollution and increasing crop yield. However, the easy volatilization and limited functional property is still the bottleneck of nitrification inhibitors. Herein, a novel water-soluble polymeric nitrification inhibitor was synthesized through the copolymerization of acrylamide and bio-based acrylic acid, which was synthesized from biomass-derived furfural, and the complexation of carboxyl groups and 3,4-dimethylpyrazole. The results showed that the nitrification inhibitor was an amorphous polymer product with a glass transition temperature of 146 °C and a thermal decomposition temperature of 176 °C, and the content of 3,4-dimethylpyrazole reached 2.81 wt%, which was 115% higher than our earlier product (1.31 wt%). The polymeric nitrification inhibitor can inhibit the activity of ammonia-oxidizing bacteria effectively, thus inhibiting the conversion of ammonium nitrogen to nitrate nitrogen and converting the insoluble phosphate into soluble and absorbable phosphate. By introducing a copolymer structure with a strong flocculation capacity, the polymeric nitrification inhibitor is further endowed with a soil-loosening function, which can increase the porosity of soil to improve the soil environment. Therefore, the nitrification inhibitor can be used in water-soluble and liquid fertilizers, as well as in high tower melting granulated compound fertilizers.

## 1. Introduction

Nitrogen utilization efficiency is notoriously poor for agricultural plants. About 50% of the N fertilizer cannot be absorbed by plants and is lost into the environment, increasing agricultural production costs and environmental pollution. One generally recognized reason is that the nitrifying bacteria can convert the ammonium nitrogen (NH4^+^-N) in soil into nitrite nitrogen (NO_2_^−^-N) and nitrate nitrogen (NO_3_^−^-N), which is less absorbable by plants. Nitrification inhibitors (NIs) are a key compound to improve fertilizer utilization efficiency and crop yield [1,2,3,4], thus reducing pollution of the soil, groundwater and atmospheric environment [5,6,7,8,9,10].

However, the commonly used NIs, such as 3,4-dimethylpyrazole phosphate (DMPP), of which the effective nitrification inhibiting ingredient is 3,4-dimethylpyrazole (DMPZ), are volatile and not resistant to high temperature [11]. Even after complexing with phosphoric acid, these shortcomings have not been effectively improved, limiting NIs’ application in some production processes that require high-temperature treatment [12,13,14,15]. Based on our earlier studies, the high temperature resistance can be improved by wrapping the NI in polymer networks, thus expanding its application in high tower melting granulation process [16,17,18,19,20]. The phosphorous-solubilizing ability can also be endowed by regulating the functional groups on the polymer networks [19,20]. 

Until now, there are no reports on the multifunctional NIs with soil-loosening abilities. Linear polyacrylamide (LPAM) is a green and environmentally friendly polymer additive to increase the porosity of soils [21,22,23,24,25,26]. It can be degraded into carbon dioxide, ammonium nitrate and water under sunlight [25,26]. When the water-soluble LPAM is applied to soils, it can immediately interact with soil particles to form soil aggregates. Thus, the soil settlement coefficient, structural coefficient and total water stabilized aggregates at all levels can be significantly improved [23,27]. This can further improve the gas and water permeabilities of soil and further prevent soil surface crusting [28,29,30]. Due to its strong flocculation capacity and less damage to the soil environment, LPAM is often used as a soil conditioner to regulate the soil environment. 

Herein, to synthesize novel NIs with high temperature resistance, effective nitrification inhibition, soil-loosening ability and phosphorus-solubilizing function, we introduce the poly(acrylamide-*co*-acrylic acid), the copolymerization product of acrylamide (AM) and acrylic acid (AA), with soil-loosening ability and phosphorus-solubilizing function into NI. The alkaline DMPZ was wrapped by the copolymer and complexed with the carboxyl functional group on the side chains of the copolymer structures. The copolymer can resist high temperature so that the volatilization of DMPZ can be prevented even under high temperature, thus improving the high temperature resistance of NI. In addition, since the poly(acrylamide-*co*-acrylic acid) is hydrophilic, the resulting NI is water-soluble. The chemical structures, thermal properties and the effect in regulating soil conditions of the water-soluble and polymeric NI (WSPNI) were characterized. The WSPNI demonstrated good soil-loosening, phosphorus-solubilizing and nitrogen fixation performances. This paper provides a new idea and method to improve the high temperature resistance and nitrification inhibition efficiency of NI and to endow NI with multifunctionality, and expand the application of NI in water-soluble fertilizers, liquid fertilizers and especially in high tower melting granulated compound fertilizers.

## 2. Materials and Methods

### 2.1. Synthesis and Materials

The synthesis procedures of WSPNI are shown in Appendix A. The detailed steps are as follows: (I) A three-neck flask containing 200 g of deionized water and a thermometer was immersed in a 45 °C water bath for 1 h. Then, 20 g of acrylamide (AM, Macklin Inc. (Beijing, China)) and 5 g of acrylic acid (AA), which was synthesized from biomass-derived furfural according to Ref. [31], were added in sequence. After being mechanically stirred for 30 min at 100 r/min, 4.5 g of self-made composite initiator was slowly added into the flask for 1 h pre-reaction; (II) 100 g of the pre-reacted products was taken out and added into another three-neck flask containing 100 g of deionized water. Then, the mechanical stirring speed and the water bath temperature were adjusted to 250 r/min and 75 °C. After the pre-reacted products were completely dissolved in deionized water, 30 g of AA, 4.5 g of self-made composite initiator and 10 g of DMPZ (Xiya Agent Co., Ltd. (Linyi, China)) were sequentially injected. After reacting for another 6 h at 75 °C with a stirring speed of 100 r/min, WSPNI with high temperature resistance and multifunctionality was synthesized. 

Herein, the self-made composite initiator used in step (I) was prepared by mixing ammonium persulfate ((NH_4_)_2_S_2_O_8_, Sinopharm Chemical Reagent Co., Ltd. (Shanghai, China)), sodium hydrogen sulfite (NaHSO_3_, Sinopharm Chemical Reagent Co., Ltd. (Shanghai, China)), sodium carbonate (Na_2_CO_3_, Sinopharm Chemical Reagent Co., Ltd. (Shanghai, China)) and azobisoheptanenitrile (AIBN, Aladdin Industrial Co., Ltd. (Shanghai, China)) with the same mass. The self-made composite initiator utilized in step (II) was prepared by mixing (NH_4_)_2_S_2_O_8_, NaHSO_3_ and AIBN with the same mass. During the synthesis procedure, the solution was alkalescent and no other agent was added for pH control.

### 2.2. Methods

The content of DMPZ in WSPNI was determined by high-performance liquid chromatography (HPLC, VERTEX-70, Ettlingen, Germany) equipped with a C18 chromatographic column. The DMPP with a purity of 97% was used as a standard sample and to obtain the linear relationship between the integrated absorption peak area and the concentration of DMPP standard aqueous solutions. Then, the HPLC graph of WSPNI aqueous solution with a concentration of 49.60 μg/mL were tested. During the test, acetonitrile/0.1‰ phosphoric acid was used as the mobile phase, and the flow rate was controlled at 0.8 mL/min. The test wavelength, column temperature, and injection volume of the WSPNI aqueous solution was set as 224 nm, 30 °C and 5 μL, respectively. 

The chemical structure of WSPNI was characterized by a Fourier transform infrared spectrometer (FTIR, Vertex-70, Ettlingen, Germany). During the test, the resolution was set as 4 cm^−1^, the number of scans was 32 and the scanning range was from 4000 to 500 cm^−1^. The molecular structure and morphology of WSPNI were characterized by an X-ray diffraction analyzer (XRD, D8 Advance, Karlsruhe, Germany). The Cu Ka with a wavelength of 1.5405 nm was chosen as the emission source. The tube voltage and current of the XRD tester were set as 40 KV and 40 mA, respectively. In addition, the scanning speed was 0.4 s, and the scanning range was from 5 to 90°. 

The thermal stability of WSPNI between 25 and 800 °C was tested by thermometric analyzer (TGA, 18 TGAQ50/DSAQ20, New Castle County, DE, USA) at a heating rate of 5 °C/min. The thermal decomposition temperature *T*_d_ was defined as the temperature at which the mass loss is 5 wt%. The glass transition temperature *T*_g_ was characterized by a differential scanning calorimetric analyzer (DSC, TGAQ50/DSAQ20, New Castle County, DE, USA). The DSC test temperature rose from 25 to 200 °C at a heating rate of 5 °C/min, then cooled down to 25 °C at a cooling rate of 5 °C/min, and finally increased to 200 °C again at the same heating rate. The first heating process was to eliminate the historic effect. All DSC data were obtained from the first cooling and the second heating processes and *T*_g_ is defined as the half-height of the corresponding step-like increase in *C*_p_. The TGA and DSC tests were both conducted under the protection of nitrogen gas. 

The phosphorus-solubilizing ability of WSPNI was verified by observing whether adding WSPNI can make a white insoluble phosphate solution undergo clarification. Herein, the white insoluble phosphate solutions were prepared by adding 5 mL of calcium nitrate solution (0.05 g/mL) and 5 mL of orthophosphate solution (pH = 8) into a beaker that contains 100 mL of deionized water. 

The soil-loosening ability of WSPNI was certificated by observing whether soil aggregates were formed after adding WSPNI into the soil solution. The detailed experiments are as follows: The uniform soil solutions were prepared by adding 50 g of fine screened soils and 200 mL of water into a measuring cylinder and then shaking. Two measuring cylinders of soil solutions were prepared of which one was added with 2 mL of WSPNI and another was used as a control. Then, the two cylinders were evenly shaken and placed side by side to observe and record the changes in soil particles and compare the settlement and aggregations of soil particles. To further demonstrate the soil-loosening ability of WSPNI, 15 g of WSPNI was applied to 10 m^2^ of soil on the campus of Qilu University of Technology, Shandong Academy of Sciences (Jinan, China). The changes in soil density and compactness at a depth of 0~10 cm before and after adding WSPNI were characterized using the cutting ring method through the soil compaction tester (TJSD-750-II, Zhejiang Top Cloud-agri Technology Co., Ltd. (Hangzhou, China)). The soil porosity was calculated from the soil density and soil specific gravity (2.65 g/cm^3^).

The nitrogen fixation ability of WSPNI is investigated by measuring the changes in the nitrification inhibition rate, the soil apparent nitrification rate and the ammonia-oxidizing bacteria (AOB) abundance under different incubation times. Since DMPZ in WSPNI is the effective component to inhibit nitrification, the nitrification inhibitory effects of WSPNI were compared with DMPP under the same DMPZ content. In the experiment, WSPNI (0.1781 g) and DMPP (0.0503 g) were dissolved in 100 mL and then separately treated with the corn-wheat rotation soil with a nitrogen content of 200 mg/kg. Then, they were cultured at 25 °C and 20% humidity for 0 h, 16 h, 32 h, 48 h, 72 h, 120 h, 168 h, 240 h, 360 h and 600 h, respectively. The changes in NH_4_^+^-N concentration, NO_3_^−^-N concentration and NO_2_^−^-N accumulation amount were determined by a destructive sampling method and continuous flow analytical system. The nitrification inhibition rate and the soil apparent nitrification rate were calculated according to the following equations. The AOB copy numbers of WSPNI- and DMPP-treated soil at 168 h, 240 h, 360 h and 600 h were determined by the real-time PCR method.
(1)$$\mathrm{Soil}\;\mathrm{apparent}\;\mathrm{inhibition}\;\mathrm{rate}=\frac{N_{1}}{\left[N_{2}+N_{1}\right]}\text{×}100\%$$
(2)$$\mathrm{Nitrification}\;\mathrm{inhibition}\;\mathrm{rate}=\frac{\left[\left(W_{1}-W_{2}\right)-\left(W_{3}-W_{4}\right)\right]}{\left(W_{1}-W_{2}\right)}\text{×}100\%$$

Herein, *N*_1_ and *N*_2_ are the content of NO_3_^−^-N and NH_4_^+^-N in the soil, respectively. *W*_1_ and *W*_3_ are the content of NO_3_^−^-N in untreated soil and NIs-treated soil before cultivation. *W*_2_ and *W*_4_ are the content of NO_3_^−^-N in untreated soil and NIs-treated soil after cultivation.

## 3. Results and Discussion

### 3.1. Content of DMPZ in WSPNI

HPLC figures of DMPP standard aqueous solutions with concentrations of 51.60, 102.04 and 130.76 μg/mL are shown in Appendix A. The absorption peak of DMPZ in DMPP standard samples appears within the time range of 8–10 min. The corresponding integral areas of DMPZ absorption peaks in Appendix A are illustrated in Table 1, which are 19.97, 39.06 and 48.63, respectively. Thus, a linear equation y = 2.6733x − 0.8473 (R^2^ = 0.9994) is obtained, which takes the integrated area of the DMPZ absorption peak as the abscissa and the concentration of DMPP standard aqueous solution as the ordinate, as shown in Figure 1.

An HPLC figure of the WSPNI aqueous solution with a concentration of 49.60 μg/mL is shown in Appendix A. An absorption peak also appears in the time range of 8–10 min, indicating the presence of DMPZ in WSPNI. The integrated area of the DMPZ absorption peak in WSPNI is 18.736. By substituting the above data into Equations (3) and (4), the content of DMPZ in WSPNI is calculated to be 2.81 wt%, which is 115% higher than our previous work (1.31 wt%) [19]. Higher DMPZ content is beneficial to inhibit the AOB activity and reduce the transformation rate of NH_4_^+^-N to NO_3_^−^-N, thus significantly reducing the application amount of N fertilizers and the input cost.
(3)$$\omega =\frac{{\rho VD\;\text{×}\;10}^{-3}}{{m\;\text{×}\;10}^{3}}\times 100\%$$
(4)$$\omega_{1}=\frac{\omega_{2}{\;\text{×}\;M}_{1}}{M_{2}}$$

Herein, *ρ, V, D* and *m* respectively represent the mass concentration of DMPP detected by the standard curve (mg/mL), the total volume of sample solution (mL), the dilution ratios of sample solution during the test, and the weight of sample (g). *ω*_1_, *ω*_2_, *M*_1_, *M*_2_ are the mass fraction of DMPZ in the sample (%), the mass fraction of DMPP (%), the molar mass of DMPZ (g/mol), and the molar mass of DMPP (g/mol), respectively [19].

### 3.2. Chemical Structures

As shown in Figure 2a, the WSPNI is a homogeneous and highly viscous material. The DSC and XRD curves of WSPNI are shown in Figure 2b,c. It can be found that WSPNI has a distinct transition from a glassy state to a rubbery state, but has no regular molecular structures. These results indicate that WSPNI is an amorphous polymer product with a high viscosity.

The FTIR spectrum of WSPNI is demonstrated in Figure 2d. The absorption peak at 3457 cm^−1^ is caused by the stretching vibration of NH_2_ and COOH. This is because COOH was not completely complexed with DMPZ, which is consistent with the result that the content of DMPZ in WSPNI is lower than the theoretical value (4.09 wt%). In addition, to better initiate the copolymerization of AA on LPAM chains, some polyacrylic acid (PAA) was also formed on the LPAM chains in the synthesis step (I), and the PAA side chains also contain COOH. The absorption peaks at 2917 and 2848 cm^−1^ are induced by the antisymmetric and symmetric stretching vibration of –CH_2_–. No out-of-plane bending vibration absorption peaks of =CH_2_ and =CH are found at 983 cm^−1^ and 818 cm^−1^, respectively, indicating that AA and AM have been completely copolymerized. The glass transition temperature of WSPNI tested by DSC is ~146 °C, which is between the glass transition temperatures of pure LPAM (~180 °C) and PAA (~80 °C), further proving that AM and AA have been copolymerized completely.

The absorption peaks at 1672 cm^−1^ and 1454 cm^−1^ are caused by C=C and C=N on the DMPZ ring, respectively, indicating that the pyrazole ring of DMPZ was not damaged in the synthesis step (II). The absorption peak at 1730 cm^−1^ is caused by C=O in COOH and CONH_2_. The two absorption peaks at 1546 cm^−1^ and 1405 cm^−1^ are caused by the asymmetric and symmetric bending vibration of COO^−^, which do not belong to DMPZ, AA or AM. This indicates that the carboxyl groups were complexed with DMPZ [32]. Based on the above discussion, the chemical structure changes in the WSPNI in the reaction steps are shown in Figure 2e.

### 3.3. Thermal Stability

As discussed in Section 3.1 and Section 3.2, DMPZ has been successfully complexed with carboxyl groups and wrapped inside the copolymer’s irregular macromolecular chains, thus preventing its volatilization and improving its temperature resistance. The thermal stability of WSPNI was tested according to Section 2.2, and its weight loss at different temperatures is shown in Figure 3. WSPNI has two weight-loss stages with weight losses of ~25 wt% (100–300 °C) and ~55 wt% (300–700 °C). The final residual carbon rate at 700 °C is ~20%. The effective decomposition temperature (*T*_d_) of WSPNI at the weight loss of 5 wt% is 176 °C, which is much higher than the volatilization temperature of DMPZ (<50 °C). Due to its high temperature resistance, WSPNI can be used as a synergistic additive for liquid fertilizers, water-soluble fertilizers, and even high tower melt-granulated compound fertilizers.

### 3.4. Nitrogen Fixation Ability

Changes in NH_4_^+^-N concentration, NO_3_^−^-N concentration, NO_2_^−^-N accumulation and total nitrogen amounts in soil under different incubation times are shown in Appendix A. The nitrification inhibition rates and the apparent nitrification rates of urea-, WSPNI- and DMPP-treated soil are respectively illustrated in Table 2. The soil nitrification inhibition rates decreased with the rise in incubation time, while the soil apparent inhibition rates increased with the rise in incubation time. The nitrification inhibition performances of WSPNI and DMPP are close since the amounts of the effective nitrification inhibitory component DMPZ are the same.

Strong nitrification occurred in the urea-treated soil. Its apparent nitrification rate sharply increased to 90.6% at 120 h, which was significantly higher than that of WSPNI- and DMPP-treated soil (*p* < 0.05). The apparent nitrification rates of WSPNI- and DMPP-treated soil at 240 h were still significantly lower than that of urea-treated soil (*p* < 0.05), and their values were close to that of the urea-treated soil at 360 h. These results indicate that WSPNI could effectively reduce the nitrification intensity.

According to the above analysis results, a strong nitrification occurred in the urea-treated soil at 168 h. Therefore, the AOB copy numbers of urea-, WSPNI- and DMPP-treated soil were investigated at 168 h, 360 h and 600 h, respectively, and the experimental results are shown in Figure 4. The copy number of AOB of urea-treated soil was significantly higher than that of WSPNI- and DMPP-treated soil (*p* < 0.05). The AOB copy number of WSPNI-treated soil showed a decreasing trend with the rise in time. However, the AOB copy number of DMPP-treated soil increased slightly at first, and then decreased at 600 h. In addition, the AOB copy number of WSPNI-treated soil was close to that of DMPP-treated soil, which is also consist with the analysis results of the soil nitrification inhibition rate. According to the aforementioned results, WSPNI can effectively inhibit the AOB activity, significantly prolong the validity period of NH_4_^+^-N and reduce the NO_3_^−^-N concentration and NO_2_^−^-N accumulation, thus reducing nitrogen losses in fertilizes. 

### 3.5. Soil-Loosening Ability

The soil-loosening ability of WSPNI is the result of the flocculation of polyacrylamide in soil. After adding WSPNI to soil, the soil particles in the test group were immediately bonded together, and thus soil aggregates containing many pores were formed, as shown in Figure 5a and Movie S1. The soil aggregates then immediately settled from the top of the cylinder to the bottom, as shown in Figure 5b,c. The reason is that the continuous accumulation of soil particles in the soil aggregates can result in a greater weight than that of the fine and dense soil particles. After settling for 135 s, the height of the finally settled soil of the test group is ~85% higher than that of the original soil before adding water, as can be seen in Movie S1, while the fine and dense soil particles of the control group was barely settled. These results show that WSPNI has a good soil-loosening ability, and the changes in the soil density, porosity and compactness before and after the addition of WSPNI are shown in Table 3. The soil density and compactness decreased by 17.4% and 14.1%, respectively, while the soil porosity increased by 18.8%. These results further demonstrate that the addition of WSPNI can form soil aggregates and thus generate plenty of pores in the soil, which is not only beneficial for improving the soil environment, but also for the growth and development of crops.

### 3.6. Phosphorous-Solubilizing Ability

The phosphorus-solubilizing processes of WSPNI are shown in Movie S2 and Appendix A. The turbid solution containing white precipitation became clarified after adding 3 mL of WSPNI and the insoluble phosphate can be converted into soluble and absorbable phosphate within several seconds. The mechanism of this process is that the metal ions in the insoluble phosphates are ionically complexed with the COOH groups on the poly(acrylmide-*co*-acrylic acid) macromolecular chains and encapsulated, thus separating them from phosphate ions. Converting a large amount of insoluble or poorly soluble phosphate into soluble phosphorus in soil not only increases the content of phosphorus that can be absorbed and utilized by plant roots, but also reduces the application amount and input cost of phosphorus fertilizers.

## 4. Conclusions

In this paper, a novel water-soluble polymeric nitrification inhibitor was synthesized through the copolymerization of acrylamide and acrylic acid and the complexation of carboxyl groups and 3,4-dimethylpyrazole. The results reveal that this nitrification inhibitor is an amorphous polymer product with a glass transition temperature of ~146 °C and thermal decomposition temperature of ~176 °C. The content of 3,4-dimethylpyrazole in the novel nitrification inhibitor is 2.808 wt%, significantly higher than our earlier product (1.306 wt%) [19]. The WSPNI can effectively inhibit the activity of ammonia-oxidizing bacteria, thus inhibiting the conversion of ammonium nitrogen to nitrate nitrogen, and can effectively convert the insoluble phosphate into soluble and absorbable phosphate. By introducing a copolymer structure with a strong flocculation capacity, the polymeric nitrification inhibitor is further endowed with soil-loosening function, which can increase the porosity of soil to improve the soil environment. Compared with other reported nitrification inhibitors [11,12,13,14,19,20], our NI possesses additional functions without the reduction in nitrification inhibition efficiency. This paper provides a new idea and method to improve the high temperature resistance of NI and to endow NI with multifunctionality. The WSPNI showed great application potential in water-soluble fertilizer and liquid fertilizer. Particularly, its high temperature resistance makes it suitable for high tower melting granulated compound fertilizer, where a temperature higher than 165 °C is usually required. In addition, the fabrication process can avoid the use of inert protective gas and organic solvent in commercial-scale production, which can reduce the production cost and improve production safety. 

## Figures and Tables

**Figure 1 polymers-16-00107-f001:**
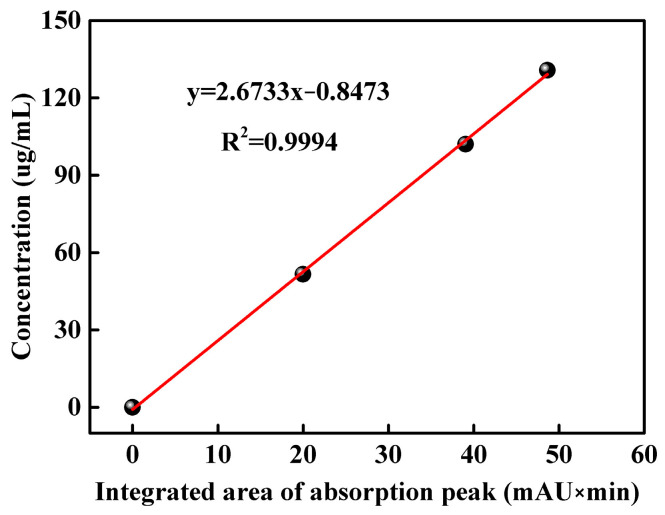
Linear relationship between the concentration of the DMPP standard aqueous solution and the integrated area of the DMPZ absorption peak.

**Figure 2 polymers-16-00107-f002:**
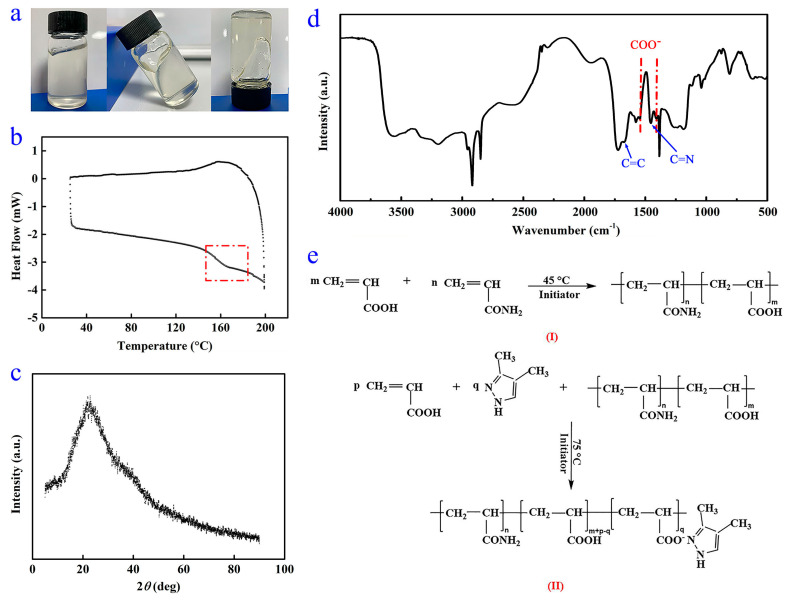
(**a**) Practical sample, (**b**) DSC curve, (**c**) XRD picture, (**d**) FTIR spectra and (**e**) changes in the chemical structure of WSPNI in reaction steps (I) and (II).

**Figure 3 polymers-16-00107-f003:**
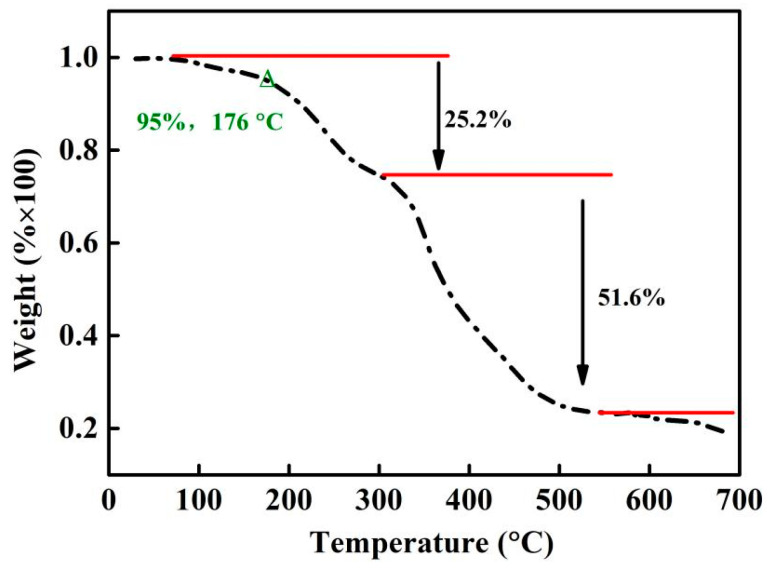
Thermal stability of WSPNI at different temperatures.

**Figure 4 polymers-16-00107-f004:**
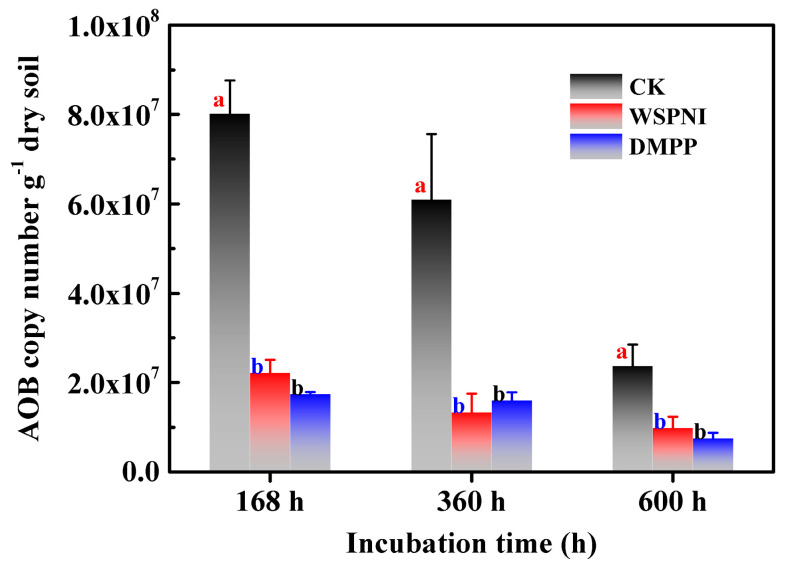
Changes in the AOB copy numbers of urea-, WSPNI- and DMPP-treated soil under different incubation times. Different lowercase letters represent statistically significant differences among treatments according to the LSD multiple range test (*p* < 0.05).

**Figure 5 polymers-16-00107-f005:**
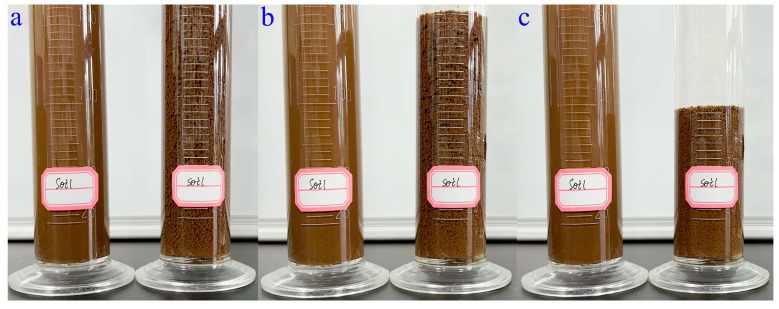
(**a**) Initial formation state, (**b**) intermediate settlement state and (**c**) final settlement state of soil aggregates after adding WSPNI into soil solution. (**Left**: reference group; **Right**: test group).

**Table 1 polymers-16-00107-t001:** The concentration of DMPP standard aqueous solutions and WSPNI aqueous solution, the areas of peaks at 8–10 min, and the calculated DMPZ content in WSPNI.

Sample	Concentration (μg/mL)	Area of Peak at 8–10 min (mAU × min)	DMPZ Content (wt%)
DMPP-std	51.60	19.97	--
DMPP-std	102.04	39.06	--
DMPP-std	130.76	48.63	--
WSPNI	49.60	18.87	2.81

**Table 2 polymers-16-00107-t002:** Nitrification inhibition rate and apparent nitrification rate of urea-, WSPNI- and DMPP-treated soil under different incubation time.

Incubation Time (h)	Nitrification Inhibition Rate (%)		Soil Apparent Nitrification Rate (%)		
	WSPNI Treated	DMPP Treated	CK (Only Urea)	WSPNI Treated	DMPP Treated
48	75.1 ± 3.02 b	85.0 ± 1.82 a	26.9 ± 2.08 a	24.9 ± 0.64 ab	20.9 ± 0.99 b
72	67.6 ± 0.31 a	68.1 ± 0.30 a	45.6 ± 0.92 a	29.9 ± 1.10 b	31.7 ± 1.18 b
120	75.6 ± 1.68 a	76.5 ± 1.62 a	90.6 ± 1.41 a	36.5 ± 1.99 b	37.4 ± 2.49 b
168	65.8 ± 0.51 b	70.0 ± 0.44 a	92.9 ± 2.09 a	51.0 ± 3.59 b	58.8 ± 3.85 b
240	48.8 ± 0.72 a	46.6 ± 0.75 a	99.2 ± 1.33 a	81.3 ± 2.12 b	76.8 ± 1.19 b
360	46.2 ± 2.49 a	41.7 ± 2.7 a	97.9 ± 0.53 a	96.6 ± 0.65 a	97.6 ± 0.61 a
600	39.9 ± 0.75 b	41.0 ± 0.74 b	99.0 ± 0.15 a	97.8 ± 0.61 b	98.1 ± 0.10 ab

Different lowercase letters represent statistically significant differences among treatments according to the LSD multiple range test (*p* < 0.05).

**Table 3 polymers-16-00107-t003:** Effects of WSPNI on soil density, compactness and porosity.

Sample	Density (g/cm^3^)	Compactness (N/cm^2^)	Porosity (%)
Without WSPNI	1.38 ± 0.04 a	417 ± 12 a	48.0%
With WSPNI	1.14 ± 0.07 a	358 ± 13 a	57.0%

The letter represents statistically significant differences among treatments according to the LSD multiple range test (*p* < 0.05).

## Data Availability

The data presented in this study are available on request from the corresponding authors.

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
