# Peer review of "Synthesizing a Water-Soluble Polymeric Nitrification Inhibitor with Novel Soil-Loosening Ability"

_polymers, 2023, doi:10.3390/polym16010107_

Round 1

Reviewer 1 Report

Comments and Suggestions for Authors

Well- presented material.

Description on the figures can be more detailed for better corelation to text and figure. For example, the FTIR peak image itself can have the peaks identified that you are pointing at.

Reviewer 2 Report

Comments and Suggestions for Authors

The research topic is important but not extremely novel. There are a few reasons mentioned below for which the manuscript in its current form may not be suitable for publication.

1)      Authors did not highlight the originality and importance of the work enough. It would have been better if authors had discussed the current work with respect to previous work in this topic and then show the novelty of the current work.

2)      The authors did not explain the design of the polymer. Authors should discuss what’s the impact of acrylic acid, acrylate, dimethylpyrazole, etc. on the overall efficiency of the polymer.

3)      Chemical structure shown in the Fig 2. Should be reviewed and corrected. Also, the figures need to be enlarged.

Comments on the Quality of English Language

The overall quality of English Language is satisfactory. Minor editing may be needed.

Reviewer 3 Report

Comments and Suggestions for Authors

This manuscript discuss about Synthesizing a Water-Soluble Polymeric Nitrification Inhibitor with Novel Soil-Loosening Ability. this paper mainly aims to endow NI with a novel soil-loosening function.

An interesting knowledge has been reported. however, the following comments should be addressed before acceptance.

The novelty, importance, and significance of the study should be more clearly mentioned in the introduction part.

How does surrounding parameter such as temperature / pH influence the synthesis of WSPNI?

Suggested to mention the functional groups in FTIR figure.

Authors should discuss more about Phosphorous-solubilizing Ability.

How does the concentration of WSPNI effects the  Nitrogen Fixation Ability?

Suggested to add future prospective in conclusion section along with obtained potential results.

Comments on the Quality of English Language

There are some typological and grammatical errors are present in the manuscript that should be revised carefully.

Reviewer 4 Report

Comments and Suggestions for Authors

The manuscript entitled “Synthesizing a Water-Soluble Polymeric Nitrification Inhibitor with Novel Soil-Loosening Ability” was well written with good results and discussion. I would like to add comments for the minor revision as follows:

1. In the Section 3.1. Content of DMPZ in WSPNI, the calculation of the content of DMPZ in WSPNI is calculated to be 2.81wt%, which is 115% higher than the previous work (1.31wt%), the Equation (S1) and (S2) should be shown in the manuscript rather than the Supplementary Materials.

2. In the Section 3.6. Phosphorous-solubilizing Ability, the author(s) should explain the phosphorus-solubilizing processes of WSPNI of how the mechanism of converting the insoluble or poorly soluble phosphate into soluble phosphorus in soil.
